# Role of PKN1 in Retinal Cell Type Formation

**DOI:** 10.3390/ijms25052848

**Published:** 2024-02-29

**Authors:** Magdalena Brunner, Luisa Lang, Louisa Künkel, Dido Weber, Motahareh Solina Safari, Gabriele Baier-Bitterlich, Stephanie Zur Nedden

**Affiliations:** Institute of Neurobiochemistry, CCB-Biocenter, Medical University of Innsbruck, 6020 Innsbruck, Austria; csat8159@student.uibk.ac.at (M.B.); luisa.lang@student.i-med.ac.at (L.L.); louisa.kuenkel@student.i-med.ac.at (L.K.); dido.weber@i-med.ac.at (D.W.); motahareh.safari@i-med.ac.at (M.S.S.); gabriele.baier-bitterlich@i-med.ac.at (G.B.-B.)

**Keywords:** protein kinase N1, NeuroD2, retinal development, retinal ganglion cells, amacrine cells

## Abstract

We recently identified PKN1 as a developmentally active gatekeeper of the transcription factor neuronal differentiation-2 (NeuroD2) in several brain areas. Since NeuroD2 plays an important role in amacrine cell (AC) and retinal ganglion cell (RGC) type formation, we aimed to study the expression of NeuroD2 in the postnatal retina of WT and *Pkn1^−/−^* animals, with a particular focus on these two cell types. We show that PKN1 is broadly expressed in the retina and that the gross retinal structure is not different between both genotypes. Postnatal retinal NeuroD2 levels were elevated upon *Pkn1* knockout, with *Pkn1^−/−^* retinae showing more NeuroD2^+^ cells in the lower portion of the inner nuclear layer. Accordingly, immunohistochemical analysis revealed an increased amount of AC in postnatal and adult *Pkn1^−/−^* retinae. There were no differences in horizontal cell, bipolar cell, glial cell and RGC numbers, nor defective axon guidance to the optic chiasm or tract upon *Pkn1* knockout. Interestingly, we did, however, see a specific reduction in SMI-32^+^ α-RGC in *Pkn1^−/−^* retinae. These results suggest that PKN1 is important for retinal cell type formation and validate PKN1 for future studies focusing on AC and α-RGC specification and development.

## 1. Introduction

The mature vertebrate retina is composed of six neuronal classes (rod and cone photoreceptors, horizontal cells, bipolar cells, amacrine cells (ACs), retinal ganglion cells (RGCs), with each containing several distinct subtypes), and three different glial cell types (Müller glia, which are the major glial cell type in mammalian retinae, microglia, and astrocytes). The axons and synaptic terminals of all these neurons are located in plexiform layers, while the cell bodies are located in nuclear layers. Photoreceptors constitute the outer nuclear layer (ONL); horizontal cells, bipolar cells, ACs, and Müller glial cells constitute the inner nuclear layer (INL); and RGCs and displaced ACs constitute the ganglion cell layer. RGCs, the only projection neurons in the retina, collect the signals and transmit them along the optic nerve, chiasm and optic tract to the target locations in retinorecipient brain areas [1]. All retinal cell classes are generated in a conserved temporal order from retinal progenitor cells (RPCs) between embryonic day (E) 11 and postnatal day (P) 10 [1], a process that is predominantly dictated by the temporal combination of various transcription factors [2]. RGCs, cone photoreceptors and horizontal cells differentiate before birth, while ACs start their development during embryogenesis but finish postnatally. Rod photoreceptors, Müller glia and bipolar cells only begin to differentiate after birth. The most diverse retinal neuronal classes are ACs, with over 60 different subtypes [3], and RGCs, with over 40 different subtypes [4,5]. Producing proper types and quantities of retinal cells constitutes the first critical step towards assembling a functional retinal circuit. Therefore, a central question in retinal development is how these different cell classes and subtypes are specified and differentiated from a common pool of multipotent RPCs.

We recently identified Protein kinase N1 (PKN1) as a developmentally active gatekeeper of the neurogenic transcription factor neuronal differentiation-2 (NeuroD2) in the cerebellum [6] as well as the hippocampus [7]. NeuroD transcription factors are known to be important regulators of retinal cell type specification [8,9,10].

During mouse retinal development, NeuroD2 is highly expressed in a subset of RGCs [8,11], glycinergic ACs, including AII ACs and in another AC cell type, likely non-GABAergic non-glycinergic ACs [3,8]. In adult retinae, NeuroD2 is also expressed in bipolar cells [8]. A loss of NeuroD2 results in a partial loss of AII ACs, and postnatal overexpression of NeuroD2 leads to increased AC formation as well as a defective stratification of AII ACs [8]. In RGCs, NeuroD2 overexpression at P0 induced RGC differentiation even after the normal embryonal developmental window of RGC genesis [8]. Besides this, the same authors reported that upon NeuroD2 overexpression, rare RGC axons were found at the optic chiasm and no axons were found in the optic tract or in retinorecipient brain areas, suggesting that RGCs overexpressing NeuroD2 might display axon pathfinding defects [8]. Indeed, cortical NeuroD2 target genes include Slit1, Robo1 and 2 and pathways that are linked to Notch and Ephrin Kinases [12], all of which are required for the correct targeting of RGC axons in the visual pathway of the brain [13].

Since PKN1 is expressed in the retina [11,14], we hypothesized that PKN1-mediated NeuroD2 suppression might also be relevant during retinal development. Therefore, the aim of this study was to analyze postnatal and adult retinae, derived from WT and *Pkn1^−/−^* mice for differences in NeuroD2 expression, RGC and AC formation as well as axon pathfinding to the optic chiasm. We report that *Pkn1^−/−^* retinae show developmental NeuroD2 overexpression, and in agreement with the literature [8], an increase in AII AC numbers. We could not see altered RGC proliferation nor defects in embryonic chiasm formation or axonal pathfinding to the optic tract. We did, however, find a specific reduction in SMI-32^+^ α-RGC in postnatal and adult *Pkn1^−/−^* retinae. Therefore, our results point towards an essential, novel role of PKN1 in AC and α-RGC formation.

## 2. Results

### 2.1. Pkn1^−/−^ Retinae Show Elevated NeuroD2 Levels

RNAscope in situ hybridization revealed that PKN1 was found throughout all retinal layers (Figure 1a). We did not observe gross differences in retinal layer thickness between postnatal (Figure 1b) or adult *Pkn1^−/−^* and WT animals (Figure 1c,d). Photoreceptor rows in the outer nuclear layer (ONL) in adult retinae were also not affected by *Pkn1* knockout (Figure 1c,e). We next chose to analyze the retinae of P10 old littermates for NeuroD2 expression, since retinal cell birth is complete around P10 in the in vivo mouse retina [15] and mouse eyes are still closed at that age. Consistent with our earlier findings in the cerebellum [6] and hippocampus [7], retinal NeuroD2 levels in postnatal P10 *Pkn1^−/−^* animals were strongly elevated compared to WT and heterozygous littermates, as analyzed by Western blotting (Figure 1f). In agreement with previous reports [8], NeuroD2 immunofluorescence staining showed that NeuroD2 was found in the RGC layer and in the INL (Figure 1g). Interestingly, we found an increased number of NeuroD2^+^ cells in *Pkn1^−/−^* retinae in the lower half of the INL, where AC cell bodies reside (Figure 1g).

### 2.2. The Effect of Pkn1 Knockout on Cells in the Inner Nuclear Layer

NeuroD2 is highly expressed in glycinergic ACs, specifically AII ACs [3,8], and Cherry et al. showed that overexpression of NeuroD2 in the postnatal retina results in enhanced AC production and abnormal stratified morphology [8]. Considering that we found more NeuroD2^+^ cells in the lower portion of the INL (Figure 1g), we hypothesized that *Pkn1^−/−^* retinae might show altered AC numbers. We stained P10 retinae from littermates for the AII AC marker disabled-1 (Dab1) and found that, indeed, *Pkn1^−/−^* retinae had an increased amount of Dab1^+^ cells, a phenotype that was also seen in adult retinae (Figure 2a, Table 1). Co-immunostaining of Dab1 and NeuroD2 revealed that most Dab1^+^ cells were also positive for NeuroD2 (Figure 2b). However, IPL stratification during development and in adult retinae, as assessed by calretinin (Figure 2c) and calbindin (Figure 2d), was similar between both genotypes. Another pan-neuronal marker, NeuN, which is expressed in horizontal cells and some amacrine cells [16], did not reveal differences between both genotypes (Appendix A, Table 1). Additionally, calbindin^+^ horizontal cells, as identified by their location in the INL, were similar in WT and *Pkn1^−/−^* retinae (Figure 2d, Table 1). Bipolar cells, stained with ceh-10 homeodomain containing homolog (Chx10), were also not affected by *Pkn1* knockout (Appendix A, Table 1). NeuroD2 overexpression has also been shown to suppress Müller glia cell production [8]; we did, however, not see any differences in the vimentin^+^ fibers in retinal sections of WT and *Pkn1^−/−^* retinae (Appendix A), suggesting no reduction in Müller glia cells. In summary, these results suggest a specific effect of PKN1 on AII AC genesis but not stratification.

### 2.3. Effect of Pkn1 Knockout on RGC and Chiasm Formation

We next analyzed the RGC layer (composed of RGC and displaced AC) of postnatal and adult retinae of both genotypes for different neuronal markers (Table 2). We found no difference in cell numbers of the RGC layer stained for calretinin, calbindin, SMI-31 or NeuN (Table 2; for representative images, refer to Figure 2c,d and Appendix A). Adult retinal flat mounts stained with the pan RGC marker RNA-Binding Protein with multiple splicing (RBPMS) also showed similar cell numbers between both genotypes (Figure 3a), suggesting no differences in RGC proliferation or RGC degeneration. Since RGCs overexpressing NeuroD2 showed a diminished axonal growth to the chiasm and no growth to the optic tract [8], we also analyzed chiasm formation in WT and *Pkn1^−/−^* embryos. RGC axons pass the optic chiasm around E12 and reach the most remote retinorecipient areas of the brain by E15 [1]; we, therefore, chose to analyze chiasm formation at E15.5. We could not see any differences in chiasm, optic nerve or optic tract thickness (Figure 3b). There was also no aberrant axonal outgrowth at *Pkn1^−/−^* chiasms, suggesting that PKN1 is not fundamental for chiasm formation.

We did, however, see a reduction in the amount of cells stained with SMI-32, a specific α-RGC marker [17], in adult *Pkn1^−/−^* retinae (Table 2). Therefore, next, we tested if this was also seen during development and if α-RGC showed altered axonal guidance to the chiasm and optic tract.

**Table 2 ijms-25-02848-t002:** Cells in the RGC layer (RGC and displaced AC) stained for various neuronal markers. All analyses were performed by an experimenter blinded to the genotype. N-values refer to different animals; unpaired *t*-tests were used for statistical comparisons ((*) *p* < 0.05). Representative calretinin and calbindin stainings are shown in Figure 2c,d. SMI-32 staining is shown in Figure 4. For NeuN and SMI-31 stainings, please refer to Appendix A, respectively.

Marker	Layer	WT Mean ± S.E.M., (n)	*Pkn1^−/−^* Mean ± S.E.M., (n)	*p*-Value
P10				
Calretinin^+^ cells/300 µm	RGC	14.4 ± 1.7 (4)	17.3 ± 1.3 (7)	>0.05
Calbindin^+^ cells/300 µm	RGC	23.1 ± 1.2 (3)	23.9 ± 0.9 (5)	>0.05
Adult				
Calretinin^+^ cells/300 µm	RGC	16.9 ± 0.9 (3)	19.9 ± 1.4 (3)	>0.05
NeuN^+^ cells/100 µm	RGC	17.4 ± 0.7 (3)	17.9 ± 1.7 (3)	>0.05
SMI-31^+^ cells/300 µm	RGC	33.5 ± 1.9 (6)	29.4 ± 2.0 (5)	>0.05
Calbindin^+^ cells/100 µm	RGC	6.4 ± 0.5 (5)	5.9 ± 0.7 (4)	>0.05
SMI-32^+^ cells/100 µm	RGC	4.5 ± 0.6 (4)	2.1 ± 0.5 (4)	<0.05 (*)

### 2.4. Effect of Pkn1 Knockout on α-RGC

Similar to our finding in adult animals (Table 2), we found a reduction in SMI-32^+^ α-RGCs in the retinal sections of the P10 old littermates (Figure 4a), pointing towards a defective cell type formation rather than a degeneration of α-RGCs upon *Pkn1* knockout. Analysis of adult retinal flat mounts stained with SMI-32 further showed that due to a significant reduction in SMI-32^+^ α-RGCs upon *Pkn1* knockout (Figure 4b–d), fewer RGC axons left the retina at the optic nerve head. The fact that the pan-RGC marker RBPMS did not reveal differences in RGC numbers is likely explained by the much lower abundance of α-RGCs (~5% of all RGCs) compared to the total amount of RGCs [18]. SMI-32^+^ α-RGC axons of *Pkn1^−/−^* animals did not show pathfinding defects to the optic nerve (Figure 4e) or optic tract (Figure 4f). This suggests a very specific effect of *Pkn1* knockout on α-RGC type formation.

**Figure 4 ijms-25-02848-f004:**
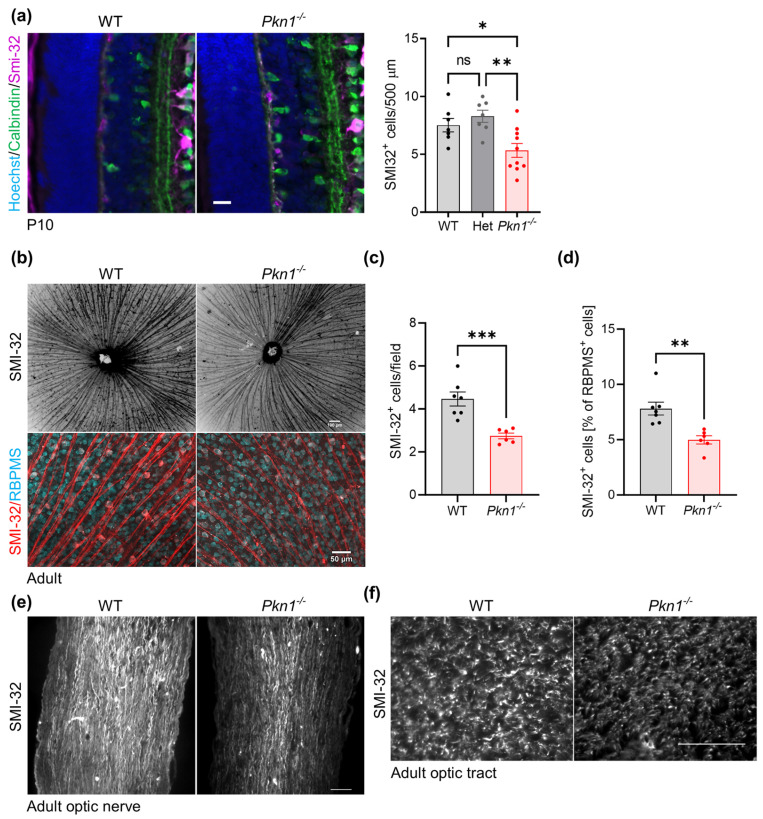
*Pkn1^−/−^* retinae show a reduced number of α-RGC. (**a**) Retinal sections prepared from P10 old littermates were stained for calbindin and SMI-32. Scale bar refers to 20 µm. SMI-32^+^ cells in the RGC layer per 500 µm were analyzed in a blinded manner (data are presented as individual n-values, referring to separate animals, with mean ± S.E.M., one-way ANOVA with Tukey’s multiple comparison test, (*) *p* < 0.05, (**) *p* < 0.01, ns not significant). (**b**) Adult retinal flat mounts were stained for SMI-32 and the pan-RGC marker RBPMS. Upper scale bar refers to 100 and lower scale bar refers to 50 µm. (**c**) The total amount of SMI-32^+^ cells (137 µm × 137 µm; data are presented as individual n-values, referring to separate animals, with mean ± S.E.M., (***) *p* < 0.001, unpaired *t*-test), and (**d**) the relative amount of SMI-32^+^ cells as a percentage of RBPMS^+^ cells per field was analyzed (137 µm × 137 µm; data are presented as individual n-values, referring to separate animals with mean ± S.E.M., (**) *p* < 0.01, unpaired *t*-test). (**e**) Optic nerve fibers were stained for SMI-32. Pictures are representative of 3 animals per genotype. Scale bar refers to 50 µm. (**f**) Optic tracts were stained for SMI-32. Pictures are representative of two animals per genotype. Scale bar refers to 50 µm.

## 3. Discussion

We found that PKN1 is widely expressed in the retina but that *Pkn1* knockout does not result in gross anatomical differences, such as altered retinal layer thickness or degeneration. However, consistent with our previous findings in the hippocampus and the cerebellum [6,7], postnatal retinal NeuroD2 levels were elevated upon *Pkn1* knockout, particularly in the INL. Although the causal involvement of NeuroD2 in the observed changes in retinal cell type formation upon *Pkn1* knockout remains to be established through in vivo knockdown/knockout of NeuroD2, we observed similarities between AC genesis in *Pkn1^−/−^* retinae and retinae with forced NeuroD2 overexpression [8]. In the postnatal INL, NeuroD2 is endogenously expressed in specific AC subtypes, such as AII AC [3,8]. Postnatal NeuroD2 overexpression results in increased AC production and defective stratification, while a loss of NeuroD2 leads to a severe reduction in AII AC [8]. In agreement with that, we found a colocalization of Dab1 and NeuroD2 immunostaining and a significant increase in AII AC in postnatal and adult *Pkn1^−/−^* retinae. However, the stratification of AC was unaffected. This suggests that *Pkn1^−/−^* retinae do not fully reciprocate ectopic NeuroD2 overexpression. A potential explanation for this discrepancy could be differences in the absolute amount of NeuroD2 levels between *Pkn1^−/−^* retinae and forced NeuroD2 overexpression induced by in vivo electroporation. There were no differences in Müller glia, horizontal cell or bipolar cell numbers, suggesting that PKN1 is specifically involved in AII AC formation.

Contrary to previous reports on postnatal NeuroD2 overexpression affecting RGC genesis [8], we could not see any differences in RGC numbers in postnatal or adult animals. It should be noted, that the effect of NeuroD2 overexpression on RGC genesis was only seen at P0 but not at P3, suggesting an active developmental time window [8]. It is likely that *Pkn1^−/−^* retinae, in analogy to the postnatal hippocampus and cerebellum show a continuous postnatal overexpression of NeuroD2 (from P1-P15) [6,7], which could result in a different effect on RGC genesis rather than overexpression induced at a certain postnatal day. In addition, even though an axon guidance defect of RGCs overexpressing NeuroD2 was reported [8], we did not see differences in optic nerve, chiasm and optic tract formation upon *Pkn1* knockout. This might also be explained by differences in absolute NeuroD2 levels between *Pkn1* knockout retinae and retinae electroporated with NeuroD2 expression plasmids [8].

We did However, see a very specific reduction in α-RGC cell numbers upon *Pkn1* knockout. α-RGC are a subgroup of RGCs, characterized by their large cell bodies, short response latency and fast conducting axons. Four different α-RGCs have been characterized, with all subtypes being stained for SMI-32 [17]. It is interesting that α-RGCs show high levels of mechanistic target of rapamycin (mTOR) signaling [19] and are very susceptible to cell death induced by Sox11 overexpression [20], a transcription factor important for RGC development [21,22].

PKN1 acts as a negative regulator of AKT signaling [6,7,23,24]; therefore, *Pkn1* knockout could have an additive effect on α-RGC mTOR signaling, which could alter α-RGC fate. Interestingly, Sox11 has also been shown to be positively regulated by NeuroD2 [25]. Therefore, future studies should be designed to analyze the role of NeuroD2 and Sox11 in α-RGC genesis upon *Pkn1* knockout.

Understanding the developmental processes that guide retinal differentiation is fundamental for the advancing of therapeutic approaches that aim to restore vision by replacing lost neurons, and our study validates PKN1 as a novel kinase involved in retinal cell type formation.

## 4. Materials and Methods

### 4.1. Animals

The generation of *Pkn1* knockout mice (*Pkn1*^−/−^ mice) has been described previously [26]. Animals were kindly provided by P. Parker (Francis Crick Institute of London, London, UK) and A. Cameron (Queen Mary University of London, London, UK). Mice were backcrossed to C57BL/6N for more than 10 generations. C57BL/6N wildtype (WT) and C57BL/6N *Pkn1^−/−^* animals were derived from the same heterozygous crosses and then bred separately but kept under the same housing and experimental conditions in the same room. C57BL/6N were derived from Jackson Laboratory. All animals in this study (younger than P12) were killed by decapitation and all adult animals were sacrificed by cervical dislocation. Male and female animals were used equally throughout this study. It should be noted that C57BL6/N animals carry the *rd8* mutation of the crumbs homologue gene 1 (*Cbr1)* gene [27]. This mutation mainly affects the outer retina at the external limiting membrane and leads to slow photoreceptor degeneration with various degrees of retinal dysplasia [28], visible from 5 weeks of age onwards [29]. Therefore, it is unlikely that the *Cbr1* mutation would affect P10 retinae, and we could not see any differences in ONL thickness between both genotypes nor did we observe abnormal lamination of the adult retina.

### 4.2. Retinal Flat Mount Staining and Analysis

Four- to twelve-week-old (adult) WT and *Pkn1^−/−^* mice were sacrificed by cervical dislocation. Eyes were enucleated and fixed in ice-cold 4% PFA, and a small hole was made in the anterior part using a needle to allow PFA to penetrate the inside of the eye. After 1.5 h of fixation, the eyes were washed in 1 × PBS and the retina was carefully isolated using a stereomicroscope (Nikon SMZ800N, Tokyo, Japan) with pointed tweezers and scissors. Isolated retinae were permeabilized with 0.3% Triton X-100 in 1 × PBS for 30 min at room temperature (RT), followed by blocking in blocking solution (10% normal goat serum (NGS), 2% BSA in 1 × PBS) for 1 h at RT. The retinae were incubated with primary antibodies (Table 3), diluted in antibody solution (0.1% Triton X-100, 5% NGS, 1% BSA in 1 × PBS + 0.005% sodium azide), for 2 days on a shaker protected from light at RT. Subsequently, the retinae were washed three times for 30 min with 0.01% Tween in 1 × PBS on a shaker protected from light at RT. Secondary antibodies (Table 3) and Hoechst (8 µM) were diluted in antibody solution, and the retinae were stained overnight on a shaker in the dark at RT. After staining, the retinae were washed 3 times for at least 30 min with 0.01% Tween in 1 × PBS on a shaker protected from light. The retinae were flattened by making 4–5 radial incisions using tweezers and scissors under the stereomicroscope and then mounted in Mowiol with the RGC layer facing upwards. Images were taken using a Zeiss Axio Imager.M2. Five pictures of each retina were taken, one at the optic nerve head and four in the peripheral regions of each retina. RBPMS and SMI-32^+^ cells were counted in five 137 µm × 137 µm squares in each of these images. The average of each picture and animal was taken and used for statistical analysis.

### 4.3. Cryosectioning

The eyes of decapitated P10 old littermates were carefully cut out and, after a small hole was made in the anterior part using a needle, fixed for 20–30 min in ice-cold 4% PFA. Care was taken not to exceed this fixation time as the antibody for NeuroD2 (see Table 3) was very sensitive to the fixation time. Adult enucleated eyes were fixed as described above (Section 4.2). After fixation, eyes were washed in 1 × PBS and dehydrated in 15% and 30% sucrose in 1 × PBS at 4 °C. Eyes were subsequently embedded in O.C.T. (CellPath (Powys, UK)) and stored at −80 °C until sectioning. Then, 15 µm thick cryosections were cut throughout the entire eye, placed onto Polysine microscope slides, and stored at −20 °C for later staining. For adult sections stained for Dab1 and Vimentin, 20 µm thick sections were analyzed. Before performing staining, cryosections were warmed to 37 °C for 1 h to assure good tissue adherence.

For immunostaining of the optic tract, a tissue block including the optic nerve, chiasm and optic tract was isolated from adult animals. The tissue was fixed in ice-cold 4% PFA, placed in 15–30% sucrose in 1 × PBS, embedded in O.C.T., and stored at −80 °C until analysis. Then, 20 µm thick tissue sections were prepared and immunofluorescence staining was performed as described below.

### 4.4. Immunofluorescence Staining of Cryosections

Cryosections were permeabilized and blocked as described above (Section 4.2), and sections were incubated with antibodies (Table 3) and diluted in antibody solution (Section 4.2) at 4 °C overnight (for Chx10 for 2 days). After washing in 1 × PBS, sections were incubated with secondary antibodies (Table 3) and Hoechst (8 µM), diluted in antibody solution, for 1.5 h at RT and protected from light. After washing three times in 1 × PBS, sections were embedded in Mowiol. Images were taken using a Zeiss Axio Imager.M2. from 5 different sections (1–3 images each) throughout the eye. For NeuroD2 and Dab1 double staining, images were taken with a confocal microscope (Zeiss LSM980, Oberkochen, Germany). FIJI/ImageJ (version 1.54g) was used for analyzing each section at 10 different spots for retinal layer thickness. For NeuroD2 expressing cells, a square of 41 µm × 41 µm was placed in 5 different INL spots (in the lower half of the INL) per section; 5 sections per animal were analyzed. The mean of each section and animal was used for statistical analysis. Cells stained with neuronal markers (Table 1 and Table 2) were analyzed per indicated length on 1 (500 µm) to 3–10 (100–300 µm) different areas of 3–5 different sections per animal. The mean of each section and animal was used for statistical analysis.

### 4.5. Isolation and Paraffin Sectioning of Optic Nerve

The optic nerve and chiasm were carefully isolated from adult animals by slowly pulling the brain from the chiasm and optic tract before cutting the optic nerves. The optic nerves were removed from the brain and fixed for 1 h in ice-cold 4% PFA. The tissue was dehydrated and embedded in paraffin. Then, 5 µm thick sections were cut on a microtome, deparaffinized and stained for SMI-32 (see Section 4.2 and Section 4.4).

### 4.6. Western Blotting

Protein extracts of whole retinae isolated from P10 old littermates were prepared in ice-cold lysis buffer (50 mM Tris pH 8.5, 1% NP-40, 5 mM EDTA, 5 mM sodium pyrophosphate, 5 mM sodium fluoride, 50 mM sodium chloride, 5 mM activated sodium orthovanadate, 30 µg/mL aprotinin, 30 µg/mL leupeptin). The retinae were additionally homogenized with a motor-driven pestle. Lysates were centrifuged at 15,000 rpm for 15 min at 4 °C, the supernatant was mixed with Laemmli sample buffer and analyzed by SDS-PAGE and Western blotting, as described previously [6]. Primary antibodies (Table 3) were added overnight in 5% BSA in TBS-T at 4 °C and secondary antibodies (LI-COR (Bad Homburg, Germany), anti-mouse 680 nm and anti-rabbit 800 nm) were added for 1–2 h, protected from light at RT. After washing in TBS-T, membranes were imaged and analyzed with an Odysee infrared Imager (LiCor).

### 4.7. In Situ RNA Hybridization

In situ hybridization was performed employing the RNAScope Fluorescent Multiplex Assay kit as per the manufacturer’s instructions (ACDBio, Newark, CA, USA) and the method reported previously [7]. Briefly, cryosections from P10 old WT retinae were dried (30 min, 60 °C) and fixed for an additional 15 min in 4% PFA. Sections were dehydrated with an alcohol serial dilution (50%, 70%, 95%, 100%) and subsequently treated with the RNA Scope^®^ 1 × Target retrieval reagents on a hot plate at 95 °C for 15 min. Protease Plus was added and incubated at 40 °C for 30 min. The pre-warmed PKN1 probe was applied and allowed to hybridize by placing the slides in an HybEZ™ Oven (2 h, 40 °C, ACDBio). After washing with PBS, the hybridization was followed by several amplification and staining steps: Amp 1 (30 min, 40 °C), Amp2 (30 min, 40 °C), Amp 3 (15 min, 40 °C), HRP-C1 (15 min, 40 °C), TSA Plus Cyanine 3 (30 min, 40 °C), HRP blocker (15 min, 40 °C), HRP-C2 (15 min, 40 °C), TSA Plus Cyanine 5 (30 min, 40 °C) and HRP blocker (15 min, 40 °C). Sections were embedded in Mowiol (Sigma-Aldrich, St. Louis, MO, USA) and imaged with a widefield microscope (Axio, Axiocam 305, Zeiss, Jena, Germany).

### 4.8. DiI/DiD Tracing

Pregnant mice (three to four months old) with E15.5 embryos were sacrificed by cervical dislocation. The abdominal wall was opened and the uterus with the embryos was removed and placed in ice-cold 1 × PBS. The embryos were removed from their amniotic sac, decapitated and the heads were fixed in ice-cold 4% PFA for 1 day at 4 °C. After fixation, heads were washed with 1 × PBS and the eyelids were carefully removed using spring scissors. The eyes were cut open and lenses were removed (but kept close to the eye) using tweezers. The eyehole was carefully dried with Whatman tissue paper. A small sticky piece of the neuronal tracer paste DiI was placed into one eyehole and the tracer DiD (NeuroTrace™ Multicolor Tissue-Labeling Kit—Invitrogen, N22884) into the other eyehole using a needle and a tweezer. After successful placing of the paste, the lenses were inserted back into the eye to keep the tracer paste in place. The heads were incubated in 1 × PBS + 0.005% sodium azide at 37 °C for 14 days. The heads were subsequently fixed with needles on a dissection dish with the anterior part facing downwards. The lower jaw and palate were carefully removed using tweezers and scissors to expose the optic nerves, the chiasm and the optic tract. Images were captured using a Zeiss Axio Imager.M2. For measuring the thickness and the length of the chiasm, the thickest and longest region in the chiasm was measured, and for analyzing the thickness of the optic nerve and the optic tract, five different regions were measured and the mean value was determined. All analyses were conducted using FIJI/ImageJ in a blinded manner.

### 4.9. Statistical Analysis

All data are presented as individual n-values with mean ± S.E.M, with n-values referring to retinae from different animals. Each figure legend clearly states if analyses were performed in a blinded manner. For comparison of two independent groups, two-tailed unpaired *t*-tests were used. For comparison of three or more groups, one-way ANOVAs with Dunnett’s or Tukey’s multiple comparison test were used. All analyses were performed in GraphPad PRISM 9.4.1.

## Figures and Tables

**Figure 1 ijms-25-02848-f001:**
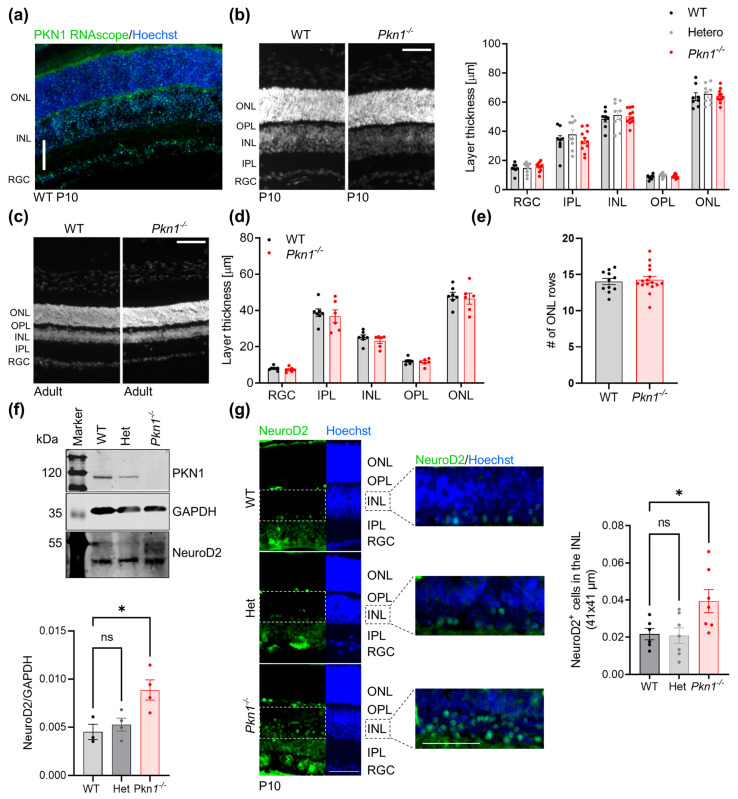
*Pkn1* knockout results in elevated developmental retinal NeuroD2 levels. (**a**) Retinal sections from P10 old WT animals were probed for PKN1 expression by RNAscope in situ hybridization. (**b**) There were no differences in retinal layer thickness between P10 old littermates (data are presented as individual n-values, referring to separate animals, with mean ± S.E.M., *p* > 0.05, one-way ANOVA). Representative image stained with Hoechst to assess layer thickness. Analysis was performed by an experimenter blinded to the genotype. (**c**) Representative image of adult WT and *Pkn1^−/−^* retinae, stained with Hoechst, to assess retinal layer thickness. (**d**) Retinal layer thickness of adult WT and *Pkn1^−/−^* retinae was not different. Analysis was performed by an experimenter blinded to the genotype (data are presented as individual n-values, referring to separate animals, with mean ± S.E.M., *p* > 0.05, unpaired *t*-test). (**e**) The number of ONL rows was not affected by *Pkn1* knockout (data are presented as individual n-values, referring to separate animals, with mean ± S.E.M., *p* > 0.05, unpaired *t*-test). (**f**) Retinal protein extracts prepared from P10 old littermates were tested for NeuroD2 expression by Western blotting. NeuroD2 levels were related to the loading control GAPDH (data are shown as individual n-values, referring to separate animals, with mean ± S.E.M., one-way ANOVA with Dunnett’s multiple comparison test, (*) *p* < 0.05, ns not significant). (**g**) Expression of NeuroD2 was analyzed in retinal sections prepared from P10 old littermates. Layers are labelled in the Hoechst channel and magnified pictures of the INL (dashed rectangles) are shown with NeuroD2 and Hoechst combined. The analysis of the number of NeuroD2^+^ cells in the INL was performed by an experimenter blinded to the genotype (data are shown as individual n-values, referring to separate animals with mean ± S.E.M., one-way ANOVA with Dunnett’s multiple comparison test, (*) *p* < 0.05, ns not significant). All scale bars refer to 50 µm. INL: inner nuclear layer; IPL: inner plexiform layer; ONL: outer nuclear layer; OPL: outer plexiform layer; RGC: retinal ganglion cell layer.

**Figure 2 ijms-25-02848-f002:**
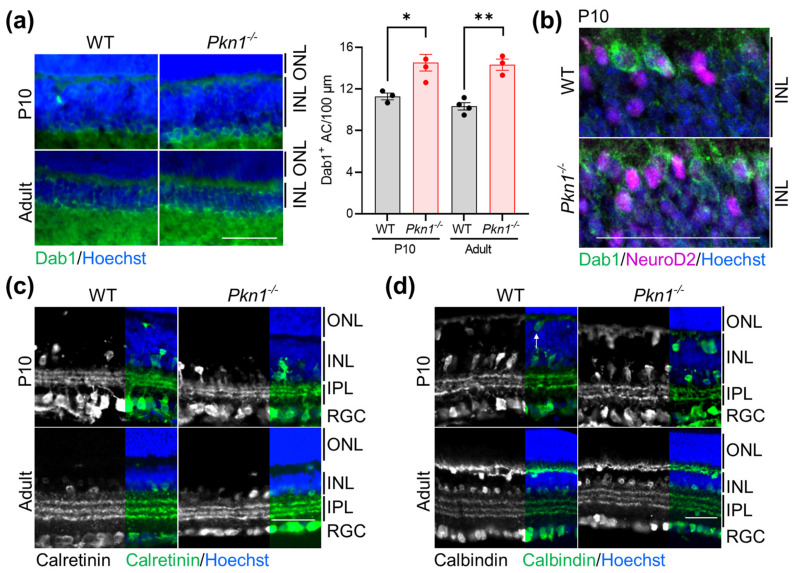
*Pkn1* knockout affects retinal amacrine cells. (**a**) Retinal sections prepared from P10 old littermates and adult animals were stained for Dab1. The number of Dab1^+^ cells in the INL was analyzed (data are presented as individual n-values, referring to separate animals with mean ± S.E.M. (*) *p* < 0.05, (**) *p* < 0.01, unpaired *t*-test). (**b**) High-magnification images of the INL of P10 retinae stained for NeuroD2 (magenta), Dab1 (green) and Hoechst. Images are representative of 3 animals/genotype. (**c**) Retinal sections prepared from P10 old littermates and from adult animals were stained for calretinin. Images are representative of 3–8 separate animals per genotype per age. (**d**) Retinal sections prepared from P10 old littermates and from adult animals were stained for calbindin. Images are representative of 3–5 separate animals per genotype per age. Arrowhead refers to horizontal cells. All scale bars refer to 50 µm. INL: inner nuclear layer; IPL: inner plexiform layer; ONL: outer nuclear layer; RGC retinal ganglion cell layer. All analyses are presented in Table 1 and Table 2.

**Figure 3 ijms-25-02848-f003:**
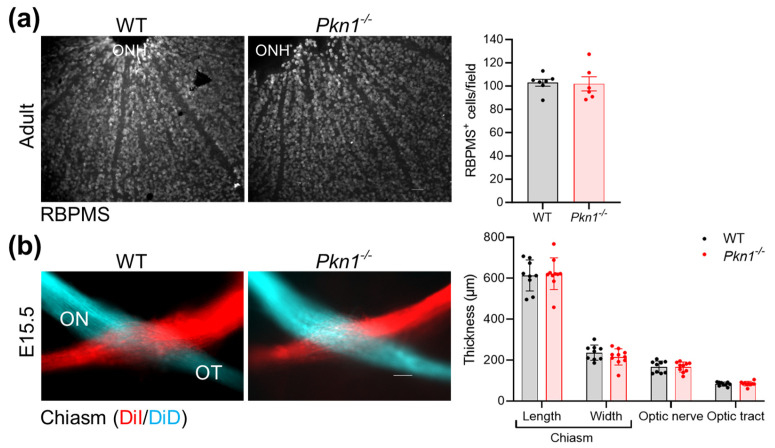
*Pkn1* knockout does not affect total RGC numbers or chiasm formation. (**a**) Total amount of RGCs, stained with the pan-RGC marker RBPMS, was not different in retinal flat mounts of WT and *Pkn1^−/−^* animals (field refers to a 137 µm × 137 µm square; data are presented as individual n-values, referring to separate animals, with mean ± S.E.M. *p* > 0.05, unpaired *t*-test). Scale bar refers to 50 µm. ONH: optic nerve head. (**b**) Chiasm formation is not affected by *Pkn1* knockout. The neuronal tracers DiI and DiD were applied to retinae of E15.5 embryos. Chiasm width, length, optic nerve (ON) and optic tract (OT) diameter were analyzed by an experimenter blinded to the genotype (data are presented as individual n-values, referring to separate animals, with mean ± S.E.M. *p* > 0.05, unpaired *t*-test). Scale bar refers to 100 µm.

**Table 1 ijms-25-02848-t001:** Analysis of cells in the INL with various neuronal markers. n-values refer to different animals; unpaired *t*-tests were used for statistical comparisons ((*) *p* < 0.05, (**) *p* < 0.01). Lower half of the inner nuclear layer (INL), HC: horizontal cells. BP: bipolar cells. Representative Dab1 and calbindin are shown in Figure 2a,b,d. Representative images for NeuN and Chx10 are shown in Appendix A, respectively.

Marker	Layer	WT Mean ± S.E.M., (n)	*Pkn1^−/−^* Mean ± S.E.M., (n)	*p*-Value
P10				
Dab1^+^ cells/100 µm	Lower INL	11.3 ± 0.3 (3)	14.5 ± 0.8 (4)	<0.05 (*)
Calbindin^+^ cells/300 µm	HC	7.1 ± 1.0 (3)	7.9 ± 0.3 (5)	>0.05
Chx-10	BP	25 ± 1.4 (3)	28 ± 2.1 (3)	>0.05
Adult				
Dab1^+^ cells/100 µm	Lower INL	10.3 ± 0.3 (4)	14.3 ± 0.5 (3)	<0.01 (**)
Calbindin^+^ cells/300 µm	HC	5.5 ± 0.6 (5)	4.3 ± 0.5 (4)	>0.05
Chx10	BP	19.1 ± 3.2 (2)	14.4 ± 1.8 (2)	>0.05
NeuN^+^/100 µm	Lower INL	21.5 ± 4.9 (3)	18.9 ± 1.4 (3)	>0.05
NeuN^+^/100 µm	HC	2.8 ± 0.4 (3)	2.9 ± 0.8 (3)	>0.05

**Table 3 ijms-25-02848-t003:** List of all antibodies used in this study. IF: immunofluorescence staining of retinal sections/flat mounts. W-Blot: Western blotting.

Antibody (Clone)	Company (Catalogue Number)	Application	Dilution
Calbindin (D1l4Q)	Abcam (ab11426) (Cambridge, UK)	IF	1:1000
Calretinin	Synaptic Systems (21411) (Göttingen, Germany)	IF	1:200
Chx10	Santa Cruz (sc-374151) (Santa Cruz, CA, USA)	IF	1:100
Dab1	Merck Millipore (AB5840-I) (Burlington, MA, USA)	IF	1:250
GAPDH (D16H11)	Cell Signaling (5174) (Danvers, MA, USA)	W-Blot	1:1000
NeuroD2 (G-10)	Santa Cruz (sc-365896) (Santa Cruz, CA, USA)	W-Blot	1:100
		IF	1:50
NeuN (D4G40)	Cell Signaling (24307) (Danvers, MA, USA)	IF	1:50–1:200
PKN1	BD Transduction Laboratories (610687) (Franklin Lakes, NJ, USA)	W-Blot	1:1000
RBPMS	Phosphosolutions (1832-RBPMS) (Denver, CO, USA)	IF	1:500
SMI-31	BioLegend (801601) (San Diego, CA, USA)	IF	1:1000
SMI-32	BioLegend (801701) (San Diego, CA, USA)	IF	1:1000
Vimentin (EPR3776)	Abcam (ab92547) (Cambridge, UK)	IF	1:250
Secondary antibodies			
goat-α-mouse IgG	Invitrogen (A11070) (Waltham, MA, USA)	IF	1:500
goat-α-rabbit IgG	Invitrogen (A21425) (Waltham, MA, USA)	IF	1:500

## Data Availability

The data presented in this study are available on request from the corresponding author.

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
