# Peer review of "Role of PKN1 in Retinal Cell Type Formation"

_ijms, 2024, doi:10.3390/ijms25052848_

Round 1
Reviewer 1 Report
Comments and Suggestions for Authors
This study explored the role of PKN1, a negative regulator of NeuroD2, during retinal development. The authors demonstrated that PKN1 knockout led to increased expression of NeuroD2 in the retina. Gross anatomy of the retina, optic chiasm formation, and axonal pathfinding defects were not observed in the PKN1 knockout mice. Meanwhile, a specific increase in Dab1+ AII amacrine cells and a reduction in SMI-32+ α-RGCs were found in Pkn-/- retinae. This manuscript is well-written and provides valuable insight into PKN1’s role in retinal development and cell type formation. Here are some minor suggestions for the authors' consideration:
1. In Figure 1g, please provide more representative images of each retinal layer to enhance clarity. It is challenging to observe the different retinal layers in the current images.
2. For Figure 2a and c, it would be beneficial for the authors to provide higher resolution or magnified images to demonstrate positive Dab1 and Calbindin staining. I would recommend arranging the representative immunofluorescence images of Dab1, Calbindin, and Vimentin in postnatal and adult mice retinae in the format of Figure 2b for better clarity.
3. Regarding Table 1 and Table 2, it could be beneficial for the authors to supply representative immunofluorescence images for each measurement.
4. Additional investigation of NeuronD2 expression in Dab1+ AII amacrine cells and SMI-32+ α-RGCs between wild-type and Pkn-/- retinae could further provide evidence of PKN1-mediated NeuroD2 suppression in different retinal cell type formation. Moreover, studies have shown PKN1 regulates NeuroD2 expression via AKT signaling. The authors could consider using a marker other than AKT for the loading control in the western blot.
Author Response
Please see attachement

Reviewer 2 Report
Comments and Suggestions for Authors
The manuscript by Brunner et al entitled “Role of PKN1 in retinal cell type formation” describes experiments and results pointing towards an essential novel role of PKN1 in AC and α-RGC formation. The manuscript is well-written, and the results are conclusive. A few grammatical mistakes and typos mentioned below need to be corrected before publication.
Minor mistakes
Line 146 ---- SMI-31
Table 2 Pkn1-/- (n=2) Authors have mentioned that 3-4 animals were used per genotype
Line 184 Pkn1 italics. Please follow gene nomenclature rules throughout the manuscript.
Line 248 RPC (RGC)
Author Response
Please see attachement

Reviewer 3 Report
Comments and Suggestions for Authors
In this manuscript, the authors explore the intriguing question of how the diverse array of retinal cell types are established. The authors present several novel findings, including altered expression of NeuroD2, SMI-32, and Dab1 in the retinae of mice lacking Pkn1, a protein kinase expressed widely in the CNS, including the retina. While much of the data are clear and the conclusions supported, the central thrust of the paper - that pkn1 acts through NeuroD2 to regulate amacrine cell and alpha-RGC differentiation - lacks support in the current form. Below I list major and minor concerns that should be addressed.
Major concerns:
- One of the key findings of the paper is that NeuroD2 expression is increased and expanded in the retinae of Pkn1 knockouts (Fig. 1). However, the immunofluorescent images presented in Fig. 1g do not look consistent with the nuclear expression pattern expected from staining of NeuroD2. In comparison to other published reports (Lin et al., Neuron 2020 PMCID PMC7529960) the staining looks different (of note, in this paper it is also not nuclear). Together, these raise concerns about specificity of the antibody. Has it been validated in knockout tissue? Are there alternative antibodies that could be used to confirm increased protein abundance?
- Related to this, my biggest concern with the paper is the attribution of all phenotypes in Pkn1-/- retinae to the up regulation of NeuroD2 (if indeed that is the case). To convincingly demonstrate that NeuroD2 is the culprit, the authors would need to inhibit NeuroD2 in the Pkn1 knockout background or do a rescue experiment to knock levels back down to wild type.
- Similarly, in Fig. 2a, the authors present data staining for Dab1 and quantify expression specifically in the ‘lower’ inner nuclear layer. It may be the resolution of my version of the paper but the staining looks very diffuse, with few cell bodies labeled. It looks quite different from other groups using this antibody, e.g. Cherry et al., 2011. How are the authors confident of the identification of individual cells?
Minor concerns:
- Line 52: ‘and’ should be inserted between ‘AC’ and ‘another’
- Table 2: formatting is a little off for the NeuN section. Why was the Dab1 quantification at P10 not included?
The writing was clear.
Author Response
Please see the attachement
